# Clinical Risk Factors and Microbiological and Intestinal Characteristics of Carbapenemase-Producing *Enterobacteriaceae* Colonization and Subsequent Infection

Wenli Yuan,[a] Jiali Xu,[a,d] Lin Guo,[b] Yonghong Chen,[c] Jinyi Gu,[a] Huan Zhang,[a] Chenghang Yang,[b] Qiuping Yang,[a] Shuwen Deng,[a] Longlong Zhang,[c] Qiongfang Deng,[b] Zi Wang,[e] Bin Ling,[b] Deyao Deng[a]

[a]Department of Clinical Laboratory, The Affiliated Hospital of Yunnan University (The Second Hospital of Yunnan Province), Kunming, Yunnan Province, China

[b]Intensive Care Union, The Affiliated Hospital of Yunnan University (The Second Hospital of Yunnan Province), Kunming, Yunnan Province, China

[c]State Key Laboratory for Conservation and Utilization of Bio-Resources, Key Laboratory for Microbial Resources of the Ministry of Education, School of Life Sciences, Yunnan University, Kunming, Yunnan Province, China

[d]Department of Clinical Laboratory, The First Affiliated Hospital of Dali University, Dali, Yunnan Province, China

[e]Department of Clinical Pharmacy, The Affiliated Hospital of Yunnan University (The Second Hospital of Yunnan Province), Kunming, Yunnan Province, China

Bin Ling and Deyao Deng contributed equally to this work.

**ABSTRACT** Gastrointestinal colonization with carbapenem-resistant *Enterobacteriaceae* (CRE) is always a prerequisite for the development of translocated infections. Here, we sought to screen for fecal carriage of CRE and identify the risk factors for CRE colonization as well as subsequent translocated pneumonia in critically ill patients admitted to the intensive care unit (ICU) of a university hospital in China. We further focused on the intestinal flora composition and fecal metabolic profiles in CRE rectal colonization and translocated infection patients. Animal models of gastrointestinal colonization with a carbapenemase-producing *Klebsiella pneumoniae* (carbapenem-resistant *K. pneumoniae* [CRKP]) clinical isolate expressing green fluorescent protein (GFP) were established, and systemic infection was subsequently traced using an *in vivo* imaging system (IVIS). The intestinal barrier, inflammatory factors, and infiltrating immune cells were further investigated. In this study, we screened 54 patients hospitalized in the ICU with CRE rectal colonization, and 50% of the colonized patients developed CRE-associated pneumonia, in line with the significantly high mortality rate. Upon multivariate analysis, risk factors associated with subsequent pneumonia caused by CRE in patients with fecal colonization included enteral feeding and carbapenem exposure. Furthermore, CRKP colonization and translocated infection influenced the diversity and community composition of the intestinal microbiome. Downregulated propionate and butyrate probably play important and multiangle roles in regulating immune cell infiltration, inflammatory factor expression, and mucus and intestinal epithelial barrier integrity. Although the risk factors and intestinal biomarkers for subsequent infections among CRE-colonized patients were explored, further work is needed to elucidate the complicated mechanisms.

**IMPORTANCE** Carbapenem-resistant *Enterobacteriaceae* have emerged as a major threat to modern medicine, and the spread of carbapenem-resistant *Enterobacteriaceae* is a clinical and public health problem. Gastrointestinal colonization by potential pathogens is always a prerequisite for the development of translocated infections, and there is a growing need to assess clinical risk factors and microbiological and intestinal characteristics to prevent the development of clinical infection by carbapenem-resistant *Enterobacteriaceae*.

**KEYWORDS** carbapenem-resistant *Enterobacteriaceae*, rectal colonization, risk factor, intestinal flora, metabolism

Address correspondence to Deyao Deng, dengdeyao2007@yeah.net.

The authors declare no conflict of interest.

Carbapenem-resistant *Enterobacteriaceae* (CRE) have emerged as a major threat to modern medicine (1, 2). Carbapenemase production is certainly the most prominent mechanism underlying these clinical CRE isolates. Commonly used antibiotics are generally inactive against CRE-associated infections. Previous studies (3–5) on the treatment of CRE infections have been particularly focused mostly on new antibiotics, including $\beta$-lactam–$\beta$-lactamase inhibitor combinations (e.g., ceftazidime-avibactam, ceftolozane-tazobactam, and meropenem-vaborbactam), and other existing antibiotics (e.g., aminoglycosides, carbapenems, and tigecycline, etc.). However, these new antibiotics have been on the market in recent years yet drug resistance still appears worldwide (6–8). Therefore, the spread of CRE is a clinical and public health problem.

Patients admitted to the intensive care unit (ICU) have been found to have a particularly high burden of CRE infections and increased mortality (9, 10). CRE can be mechanistically classified into carbapenemase-producing *Enterobacteriaceae* (CPE) and non-carbapenemase-producing carbapenem-nonsusceptible *Enterobacteriaceae*. Three groups of carbapenemases, KPC-2, NDM, and VIM, are currently considered to be the three major $\beta$-lactamases of epidemiological and clinical significance (11). Knowing the carbapenemase type is vital for predicting the utility of antibiotics. Ceftazidime and aztreonam retain activity against CPE with OXA-48-like enzymes as long as these strains do not also produce AmpC or extended-spectrum $\beta$-lactamase (ESBL) enzymes (12, 13), ceftazidime-avibactam covers *Enterobacterales* with KPC or OXA-48-like carbapenemases *in vitro* (14, 15), and aztreonam-avibactam should additionally cover *Enterobacterales* with NDM (16). In the last decade, the therapeutic recipe for CRE infections might be targeted and personalized based on molecular resistance phenotypes, disease severity, and patient characteristics (17–20). It is essential that clinical microbiology laboratories be capable of recognizing CRE strains that produce these key groups of carbapenemases and refer them for further testing when appropriate to inform clinicians and infection preventionists.

Sites of CRE carriage include the lower gastrointestinal (GI) tract, oropharynx, skin, urine, and the intestine are the most likely endogenous reservoirs (21–23). *Enterobacteriaceae*, including *Klebsiella pneumoniae*, possibly reside as colonizers in the human intestine. The prevalence of CRE colonization in hospitalized patients ranges from 3 to 18% (23–25); however, it can be higher in critically ill patients (26). Recently, epidemiological studies (27–30) have suggested that the majority of *K. pneumoniae* infections are preceded by the colonization of the gastrointestinal tract, and in ICU patients colonized with CRE, the risk of systemic CRE infection varies widely, between 29 and 73%. Our formal investigation (28) and another study (29) showed that *K. pneumoniae* carriage of the $bla_{OXA-48-like}$ gene has become endemic in fecal colonization in hospitals. In clinical practice, there is a growing need to assess the impact of prior colonization by carbapenem-resistant *K. pneumoniae* (CRKP) on new systematic infections. Therefore, we hypothesize that fecal carriage of CRE carrying the $bla_{OXA-48-like}$ gene probably contributed to the increasing prevalence of clinical infections by such carbapenemase producers.

Colonization of the intestine by nonhost niche microorganisms can lead to a permeable gut through various mechanisms (30–38), by either being directly responsible for the features of inflammation (30–32) or favoring microbe-microbe and microbe-host interactions such as regulating the diversity and community composition of the cecal microbiome (33, 34), manipulating bacterial metabolites (35, 36), participating the innate and the adaptive immune responses (37), and penetrating host barriers (38), allowing colonizing microorganisms to invade the intestinal tissue and leading to subsequent translocated infection.

In the current study, we investigated the fecal carriage of CRE and the carbapenemase genotypes and identified independent risk factors for CRE colonization and subsequent translocated pneumonia in critically ill patients admitted to the ICU of a university hospital in China. The intestinal flora composition and fecal metabolic profiles were also determined. To our knowledge, there is scarce information about whether identified CRE rectal carriers are prone to having subsequent infection with CRE. To test the hypothesis, we established animal models gastrointestinally colonized with a CRKP clinical isolate expressing green fluorescent protein (GFP) in C57BL/6J mice and traced the subsequent systemic infection. The intestinal barrier, inflammatory factors,

**TABLE 1** Multivariate analysis of risk factors for CRE colonization[a]

| Risk factor | P value (OR [95% CI]) |
|---|---|
| APACHE II score of ≥21 | 0.046 (1.328 [1.037–3.914]) |
| Carriage of or infection by another MDRO | <0.001 (9.555 [3.628–25.165]) |
| No. of days of hospitalization | 0.012 (1.092 [1.019–1.170]) |
| Serum albumin | 0.005 (0.854 [0.764–0.954]) |

[a]MDRO, multidrug-resistant organism (i.e., VRE, MRSA, and ESBL-carrying organisms); APACHE, acute physiology and chronic health evaluation.

and infiltrating immune cells were further investigated in colon tissues collected from CRKP-colonized and translocated infection models. Our objective was to identify the risk factors and intestinal biomarkers for subsequent infections among CRE-colonized patients, which can be used to control these factors and to direct empirical antimicrobial therapy when necessary.

## RESULTS

**Characteristics of the ICU patient cohort.** The study enrolled 374 patients who were screened for fecal colonization at the time of ICU admission. Study participants were predominantly male ($n = 210$; 56.1%), with an average age of 62 years (range, 16 to 95 years). There were 54 patients identified as CRE rectal carriers. Among them, 27 (50%) developed a subsequent CRE-positive culture from the respiratory tract, which was interpreted as pneumonia by clinicians. As summarized in Table 1, patients colonized with CRE were significantly more likely than noncolonized patients to have a long hospital stay, lower serum albumin levels, higher acute physiology and chronic health evaluation II (APACHE II) scores, and easier carriage of or infection with another multidrug-resistant organism (MDRO) in the hospital. Overall, the colonized and noncolonized groups did not significantly differ by age, gender, hospitalization within the past 6 months, and antibiotic exposure in the hospital, except for carbapenems (see Table S2 in the supplemental material).

**Microbiological characteristics.** As shown in Table 2, *Klebsiella pneumoniae* ($n = 50$; 92.6%) was the predominant CRE species identified by rectal screening. Other CRE isolates included *Escherichia coli* ($n = 2$; 3.7%) and *Enterobacter cloacae* ($n = 2$; 3.7%). Of 54 CRE colonizing isolates, 52 were identified as CPE by Carbapenemase Nordmann-Poirel (CarbaNP) and modified carbapenem inactivation method (mCIM) with or without an EDTA-modified carbapenem inactivation method (eCIM) test. A total of 51 feces-colonizing CRE isolates were positive for the various carbapenemase genes screened, and $bla_{OXA-48-like}$, $bla_{KPC-2}$, and $bla_{NDM-1}$ genes were detected in these isolates. The $bla_{OXA-48-like}$ (24/51; 47%) and $bla_{KPC-2}$ (23/51; 45%) genes were the predominant carbapenemase genes, followed by $bla_{NDM-1}$ (4/51; 8%). Twenty-four isolates harboring the $bla_{OXA-48-like}$ gene all belonged to sequence type 231 (ST231), and the major ST of *Klebsiella pneumonia* isolates carrying $bla_{KPC-2}$ was ST11 (20/22; 90.9%). Compared with the colonizing CRE isolates, CPE bacteria (26 *Klebsiella pneumonia* isolates and 1 *Enterobacter cloacae* strain) isolated from respiratory tract specimens were identified with the same carbapenemase gene and ST type. As expected, the

**TABLE 2** Microbiological characteristics of carbapenemase-producing *Enterobacteriaceae* in patients with fecal colonization

| Carbapenemase gene | Bacterium (no. of isolates) | ST(s) |
|---|---|---|
| $bla_{OXA-48-like}$ | *Klebsiella pneumoniae* (24) | ST231 |
| $bla_{KPC-2}$ | *Klebsiella pneumoniae* (22) | ST11, ST258,[a] ST690[a] |
| | *Enterobacter cloacae* (1) | ST131[b] |
| $bla_{NDM-1}$ | *Escherichia coli* (2) | ST88 |
| | *Klebsiella pneumoniae* (1) | ST11 |
| | *Enterobacter cloacae* (1) | ST509 |

[a]One strain with the indicated ST.
[b]ST131 and ST88 are interchangeable.

**TABLE 3** Cox regression analysis of clinical variables associated with subsequent pneumonia caused by carbapenemase-producing *Enterobacteriaceae* in patients with fecal colonization

| Clinical variable | Univariate analysis | | Multivariate analysis | |
|---|---|---|---|---|
| | Hazard ratio (95% CI) | P value | Hazard ratio (95% CI) | P value |
| Enteral feeding | 3.942 (1.585–9.805) | 0.003 | 3.779 (1.508–9.469) | 0.005 |
| Nasogastric tube | 2.788 (1.119–6.945) | 0.028 | | |
| Carbapenem exposure | 3.744 (1.622–8.643) | 0.002 | 3.646 (1.547–8.594) | 0.003 |
| PPI use | 2.629 (1.109–6.228) | 0.028 | | |
| Total serum protein concn of ≤60 g/L | 0.388 (0.174–0.865) | 0.021 | | |

production of OXA-48-like carbapenemase (14/27) was the main antibiotic resistance mechanism, followed by KPC-2 (12/27) and NDM-1 (1/27). ST231 isolates harboring $bla_{OXA-48-like}$ and ST11 isolates harboring $bla_{KPC-2}$ were the most prevalent isolates (Table 2).

**Clinical outcomes.** The highest rate of all-cause in-hospital mortality was recorded in fecally colonized patients with subsequent translocated CRE-associated pneumonia (17 of 27; 63.0%); a lower rate was documented in rectal carriers (25 of 54; 46.3%), whereas the lowest rate was recorded in noncolonized patients (102 of 320; 31.9%). Of 54 CRE fecally colonized patients, 27 (50%) developed CRE-associated pneumonia in the hospital, and the median time from admission to the onset of infection was 4 days (interquartile range [IQR], 2 to 11 days). Of the 117 CRE isolates associated with pneumonia cases in this study, 27 belonged to the CRE-colonized group and 90 belonged to the noncolonized group. The characteristics of CRE-associated pneumonia are shown in Table 3. Upon multivariate analysis, additional variables associated with subsequent pneumonia caused by CRE in patients with fecal colonization included the following factors: enteral feeding and carbapenem exposure.

**CRKP colonization and translocated infection influence the diversity and community composition of the gut microbiome.** In order to evaluate the effect of CRKP colonization and translocated infection on the diversity and community composition of the intestinal microbiome, the gut microbiota of the three groups were investigated based on 16S rRNA sequencing. Fecal samples collected from CRKP-colonized patients showed dramatic alterations in the gut flora composition compared with the control group. In the control group, the dominant phyla were *Firmicutes* (40.87%), *Bacteroidota* (30.68%), *Proteobacteria* (9.87%), and *Actinobacteriota* (3.96%). However, CRKP colonization challenge significantly changed the composition of bacteria at the phylum level: the relative abundance of *Proteobacteria* increased to 17.50%, and the relative abundances of *Bacteroidota* and *Actinobacteriota* decreased to 20.42% and 0.50%, respectively (Fig. 1a and b). Alpha diversity was evaluated by Chao1 and Shannon indices. After CRKP colonization, the Chao1 index increased compared with the microbiota from the control group, but the Shannon index showed no statistical difference (Fig. 1c). Beta diversity results suggested that the gut microbiota of CRKP-colonized patients did not show a significant difference from the control group (data not shown). A histogram of linear discriminant analysis (LDA) scores analyzed by the LDA effect size (LEfSe) algorithm was drawn to identify statistically significant biomarkers and reveal the dominant microorganisms in each group. As shown in Fig. 1d, dominant communities of 7 and 11 taxa were found in the control and CRKP rectal colonization groups, respectively.

In this experiment, we further determined the gut microbiota diversity of the CRKP colonization group and the with translocated infection group. Operational taxonomic units (OTUs) for each group were evaluated to identify the unique and shared genera. A Venn diagram (Fig. 1e) displaying the overlap between the two groups showed that 131 OTUs were shared. A total of 327 OTUs were unique to the CRKP intestinal colonization group, and 91 OTUs were unique to the CRKP translocated infection group. After CRKP translocated infection, the Chao1 index was significantly increased compared with the microbiota from the colonization group, but the Shannon index and beta diversity still showed no statistical differences (Fig. 1f). The gut microbiota community compositions of the CRKP colonization

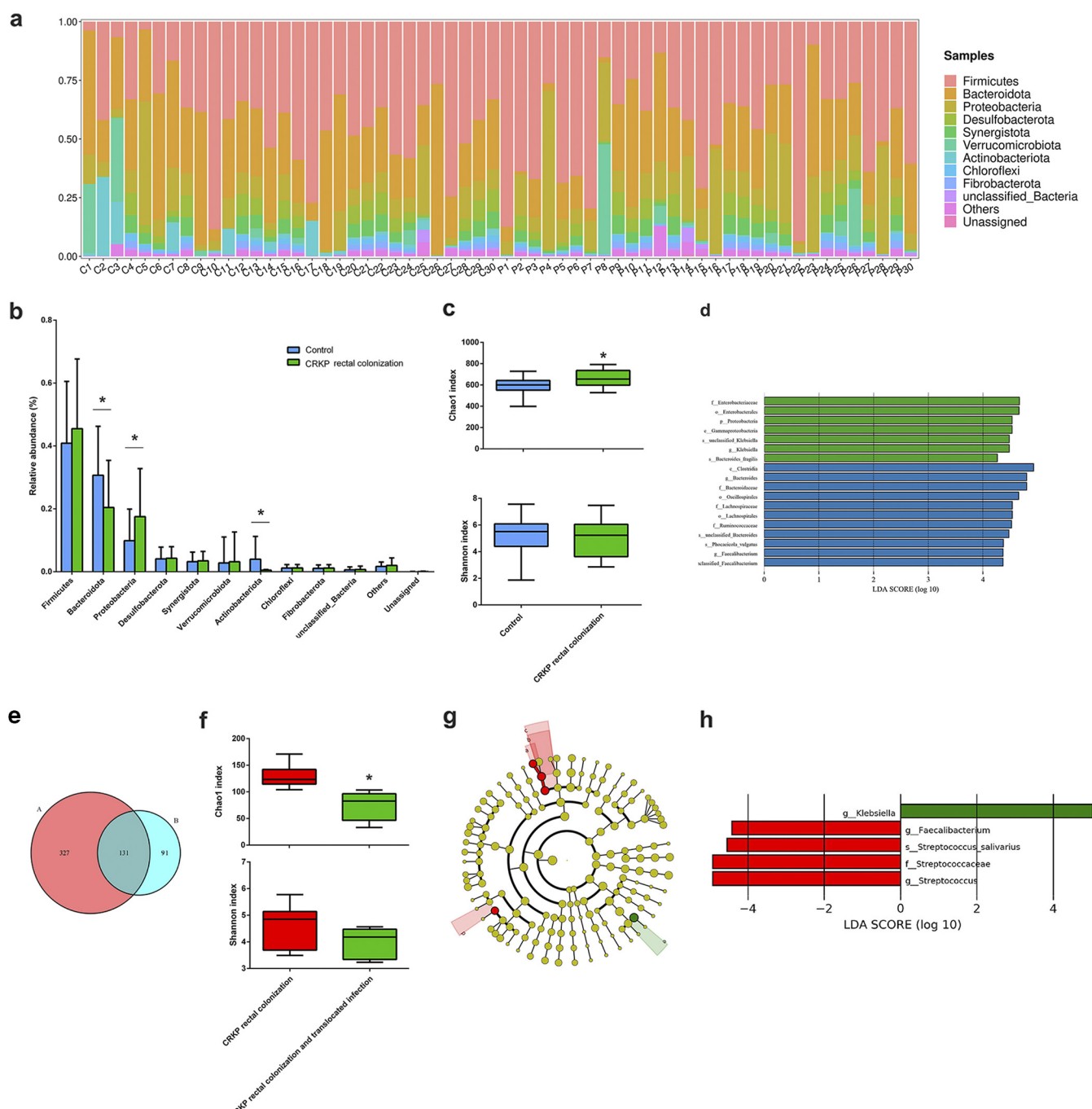

**FIG 1** CRKP colonization and translocated infection influence the intestinal flora composition. (a) Compositional levels of fecal flora in the CRKP colonization group ($n$ = 30) (C1 to C30) and the control group ($n$ = 30) (P1 to P30) at the phylum level. (b) Statistics of the changes in relative abundance induced by CRKP colonization. (c) Alpha diversity indicated by the Chao1 index and Shannon index (*, $P$ < 0.05 versus the control). (d) Distribution histogram based on linear discriminant analysis (LDA). The default criteria of an LDA score of >4 and a $P$ value of <0.05 indicate different species and a higher abundance in one group than in the other. The histogram of LDA scores calculated for selected taxa shows significant differences in microbe types and abundances between the control group (blue) and the CRKP colonization group (green). LDA scores on a $\log_{10}$ scale are indicated at the bottom. The significance of the microbial marker increases with the LDA score. (e) Venn diagram displaying the operational taxonomic unit (OTU) overlap between the CRKP colonization group (A, left, pink) ($n$ = 6) and the translocated infection group (B, right, powder blue) ($n$ = 6). (f) Alpha diversity indicated by the Chao1 and Shannon indices between the CRKP colonization group ($n$ = 6) and the translocated infection group ($n$ = 6) (*, $P$ < 0.05 versus the CRKP colonization group). (g) Linear discriminant analysis effect size (LEfSe) and LDA based on OTU characterizations of the microbiota of the CRKP colonization group ($n$ = 6) and the translocated infection group ($n$ = 6). The cladogram generated by the LEfSe method shows the phylogenetic distribution of fecal microbiomes associated with the two groups. Each filled circle represents one phylotype. The circle size is proportional to the phylotype abundance. By default, the taxonomic levels are arranged outward from phylum to genus. Red circles on the branches represent microbial communities playing pivotal roles in CRKP rectal colonization. Green circles represent microbial groups playing important roles in CRKP translocated infection. The default criteria of an LDA score of >4 and a $P$ value of <0.05 indicate different species and a higher abundance in one group than in the other. a, s_*Streptococcus*_salivarius; b, g_*Streptococcus*; c, f_Streptococcaceae; d, g_*Faecalibacterium*; e, g_*Klebsiella*. (h) Histogram of LDA scores calculated for selected taxa showing significant differences in microbe types and abundances between the CRKP colonization group (red) and the translocated infection group (green). LDA scores on a $\log_{10}$ scale are indicated at the bottom. The significance of the microbial marker increases with the LDA score.

**TABLE 4** Results of statistical analysis of the main metabolites changed in feces for CRKP intestinal colonization versus CRKP translocated infection ($n = 6$)

| Metabolite | P value | Fold change | VIP score | Trend |
|---|---|---|---|---|
| Asparaginyl-hydroxyproline | 0.008 | 4.577 | 2.627 | Up |
| Gyromitrin | 0.023 | 1.682 | 2.219 | Up |
| Ethyl hexadecanoate | 0.027 | 2.577 | 2.305 | Up |
| 5-(2-Furanyl)-3,4-dihydro-2H-pyrrole | 0.029 | 1.873 | 2.167 | Up |
| 2-Methyl-1,3-cyclohexadiene | 0.029 | 3.331 | 2.264 | Up |
| Sciadonic acid | 0.031 | 0.502 | 2.292 | Down |
| 5-(3′,4′-dihydroxyphenyl)-$\gamma$-valerolactone-3′-O-glucuronide | 0.031 | 0.347 | 2.229 | Down |
| Armillane | 0.039 | 1.500 | 2.089 | Up |
| Prolyl-$\gamma$-glutamate | 0.041 | 3.154 | 2.204 | Up |
| 1-Methylhistamine | 0.041 | 2.724 | 2.177 | Up |
| Pyrocatechol | 0.043 | 3.913 | 2.090 | Up |
| 1,2,3,4-Tetrahydro-1,5,7-trimethylnapthalene | 0.048 | 2.237 | 2.003 | Up |
| $\beta$-D-Glucosamine | 0.049 | 2.068 | 2.125 | Up |

group and the translocated infection group were further analyzed by the LEfSe method. Dominant communities of four taxa and one genus were found in the CRKP colonization group and the translocated infection group, respectively. Among them, *Klebsiella* (genus) was dominant in the CRKP translocated infection group, and *Streptococcus salivarius* (species), *Streptococcus* (genus), *Streptococcaceae* (family), and *Faecalibacterium* (genus) were dominant in the CRKP colonization group (Fig. 1g). The relative abundances of selected taxa showed that the abundances of *Streptococcus salivarius*, *Streptococcus*, *Streptococcaceae*, and *Faecalibacterium* were significantly decreased in the CRKP translocated infection group, but the abundance of *Klebsiella* (genus) was significantly increased (Fig. 1h).

**CRKP colonization and translocated infection influence fecal metabolic profiles.** To investigate whether CRKP colonization and translocated infection induce fecal metabolic disorders, an ultraperformance liquid chromatography-tandem mass spectrometry (UPLC-MS/MS) assay was used to analyze nontargeted metabolomes in fecal samples collected from CRKP rectally colonized patients. As shown in Table 4, 13 altered metabolites in feces were obtained. However, no correlation with microorganism biomarkers in the intestinal flora of CRKP translocated infection patients was observed. We further observed short-chain fatty acid (SCFA) concentration differences among the control group, the CRKP rectal colonization group, and the translocated infection group. Downregulated propionate and butyrate were observed in fecal samples collected from CRKP rectally colonized patients compared with the control group, especially in the CRKP translocated infection group (Fig. 2).

**CRKP colonization and translocated infection disrupt the intestinal barrier.** Mice were euthanized and imaged usings the Lumina series III *in vivo* imaging system (IVIS) to observe CRKP colonization of the gastrointestinal tract and subsequent translocated infection. As shown in Fig. 3, CRKP-associated pneumonia in colonized mice was observed approximately 15 days after CRKP intestinal colonization. Hematoxylin and eosin (HE) staining was used to evaluate histopathological injury. The colons of control mice had intact mucous membranes and healthy crypt structures (Fig. 4a). However, the mice in the CRKP-colonized group showed intestinal mucosa and submucosal edema, inflammatory cell infiltration, and epithelial injury. To evaluate the influences of CRKP rectal colonization and translocated infection on the intestinal barrier, tight junction (TJ) proteins and mucin2 (MUC2) were measured in colon tissues. The contents of occludin, ZO-1, claudin-3, and MUC2 were analyzed by an enzyme-linked immunosorbent assay (ELISA), and immunofluorescence staining of TJ proteins was further performed. As shown in Fig. 4, CRKP intestinal colonization resulted in obviously decreased concentrations of TJ proteins and MUC2, whereas CRKP-colonized mice challenged with subsequent translocated infection displayed the lowest concentrations of TJ proteins. However, no significant differences in MUC2 concentrations in colon tissues were observed after translocated infection challenge. The immunofluorescence intensity of TJ proteins was also reduced in the colon tissues of mice with CRKP gut colonization, especially in mice with translocated infection (Fig. 4b).

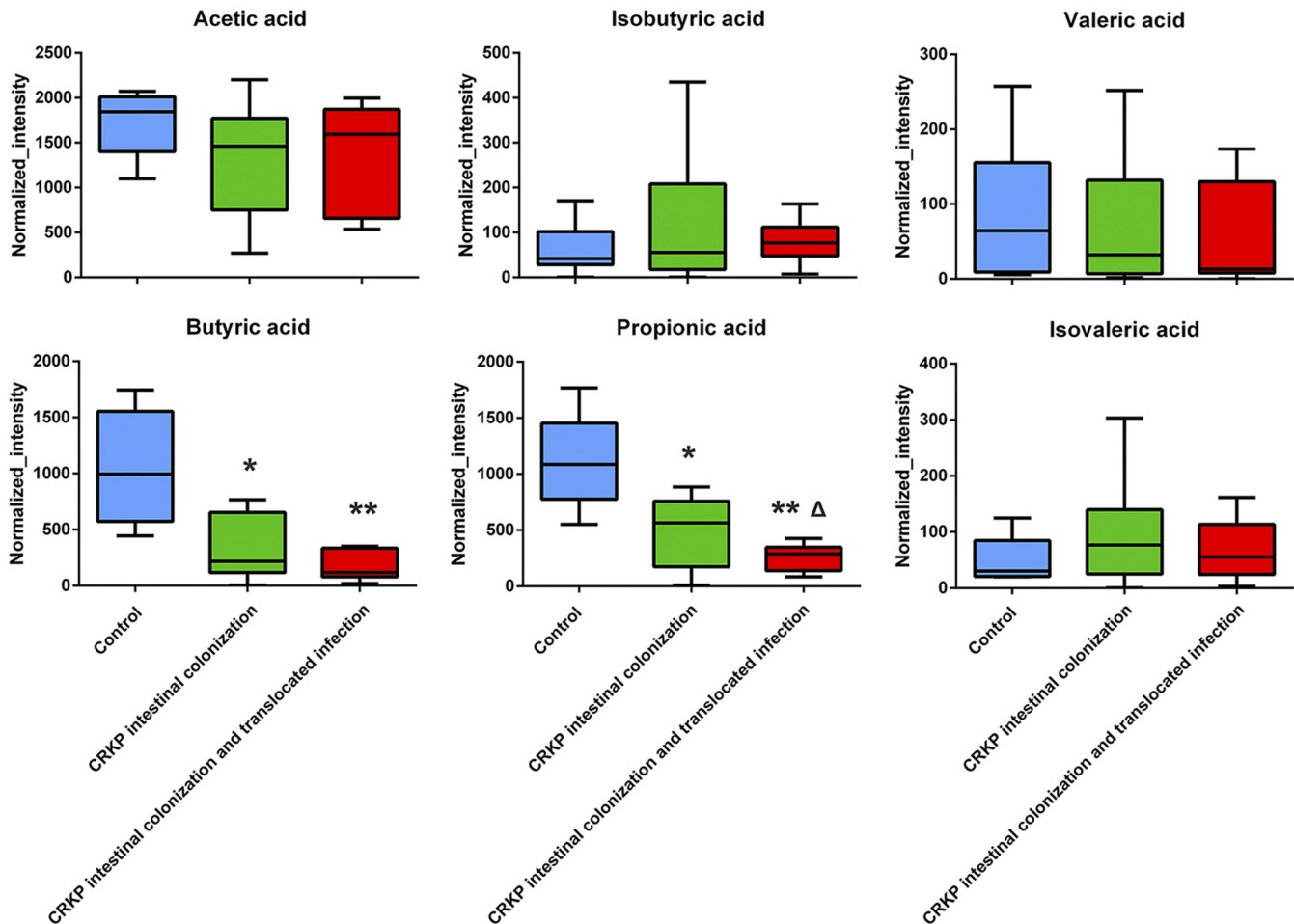

**FIG 2** Results of statistical analysis of short-chain fatty acid metabolites changed in feces ($n = 9$) (**, $P < 0.001$ versus the control group; *, $P < 0.05$ versus the control group; Δ, $P < 0.05$ versus the CRKP rectal colonization group).

**CRKP colonization and translocated infection induce obvious immune cell infiltration and inflammatory factor expression.** To determine the features of immune cell infiltration in colon tissues challenged with Actinobacteria CRKP colonization and translocated infection, we performed double-immunofluorescence staining for each group. As shown in Fig. 5a, CD3$^+$ CD8$^+$ lymphocytes, CD19$^+$ CD20$^+$ lymphocytes, and CD80$^+$ CD86$^+$ macrophages were all observed in the three groups, especially in mice colonized with CRKP. Importantly, more CD80$^+$ CD86$^+$ macrophages existed in colon tissues harvested from CRKP-colonized mice with translocated infection. However, translocated infection did not obviously enhance the numbers of CD3$^+$ CD8$^+$ lymphocytes and CD19$^+$ CD20$^+$ lymphocytes in CRKP-colonized mice (Fig. 5b). In order to evaluate the influences of CRKP colonization and translocated infection on inflammatory factor expression, interleukin-1$\beta$ (IL-1$\beta$), tumor necrosis factor alpha (TNF-$\alpha$), IL-6, and IL-10 concentrations were analyzed. The levels of inflammatory factors in the colon tissues of CRKP colonization-challenged mice were significantly higher than those in normal mice, with significantly enhanced TNF-$\alpha$, IL-1$\beta$, and IL-6 expression, and translocated infection further enhanced the expression of IL-6 (Fig. 5c). Moreover, the concentration of the anti-inflammatory cytokine IL-10 was slightly decreased compared to the levels in the control group.

## DISCUSSION

CRE infections are associated with considerable mortality, with rates of up to 50% being reported in the literature (39). Our study found that the all-cause in-hospital mortality rate

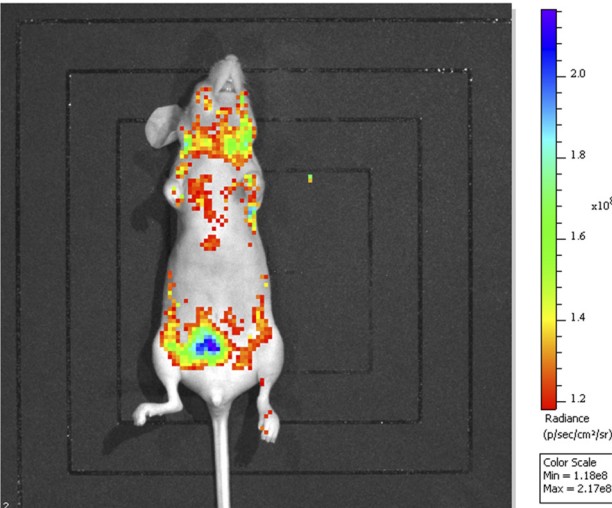

**FIG 3** *In vivo* evaluation of CRKP colonization of the gastrointestinal tract and subsequent infection of C57BL/6J mice with immunosuppression and PPI use. Live-animal and *ex vivo* IVIS imaging confirmed CRKP colonization of the gastrointestinal tract and subsequent infection. The IVIS images were collected 25 days after oral gavage of GFP-labeled CRKP, and the nonspecific distribution of CRKP in multiple organs such as the lung, heart, and brain, in addition to the intestinal tract, is shown.

was 46.3% in rectal carriers; however, 50% of colonized patients probably developed CRE-associated pneumonia in the hospital. Most importantly, these patients displayed a significantly high mortality rate (63.0%). There are many contributors to this high rate of mortality, including longer hospital stays, poorer overall health status, systematic infection, comorbid illnesses, and limited antimicrobial options for treating these infections. Our results also emphasize the importance of recognizing and eliminating the asymptomatic carrier state to prevent the development of clinical infection with CRE, such as rectal decontamination therapy with gentamicin or colistin (40).

In order to identify and prevent intestinal tract colonization in clinical practice, investigations of the clinical risk factors for rectal CRE colonization should not be delayed, especially in critically ill patients in ICU wards. Upon multivariate analysis, an APACHE II score of ≥21, carriage of or infection with another MDRO, length of hospitalization, and serum albumin might be independent risk factors for fecal CRE colonization in our institution. It is well known that the length of hospitalization is a high-risk factor for CRE colonization or infection (41). A matter of concern is that patient carriage of or infection with another MDRO has been shown to increase the risk of CRE colonization more than 9-fold (42). As in this and other studies (35, 36), CRE colonization is commonly associated with low SCFA levels. Moreover, these SCFAs serve to enhance the mucosal barrier and inhibit intestinal inflammation (30–32). It is also possible that established CRE colonization perturbs the abundance of MDROs in the intestinal microbiota by decreasing the SCFA levels, which in turn may serve as a pathogenic community to preserve host pathology. Our study suggested that infection with another MDRO most likely could be both a cause and a result of CRE colonization.

In our investigation, no CPE isolates coharboring carbapenemases were identified. This study observed 24 CPE isolates harboring the $bla_{OXA-48-like}$ gene, and clone ST231 was the prevalent ST. In contrast to previous studies (28), we found that the $bla_{OXA-48-like}$ gene was the predominant carbapenemase gene, and all 24 $bla_{OXA-48-like}$ gene-carrying CPKP isolates screened were identified exclusively as clone ST231. Notably, these bacteria were isolated at different times in the hospital, so the possibility of a nosocomial infection outbreak can probably be ruled out. An explanation for this high rate is that these bacteria were isolated during rectal screening. Clinical CPE isolates with OXA-48-like enzymes always displayed high-level resistance to temocillin and piperacillin-

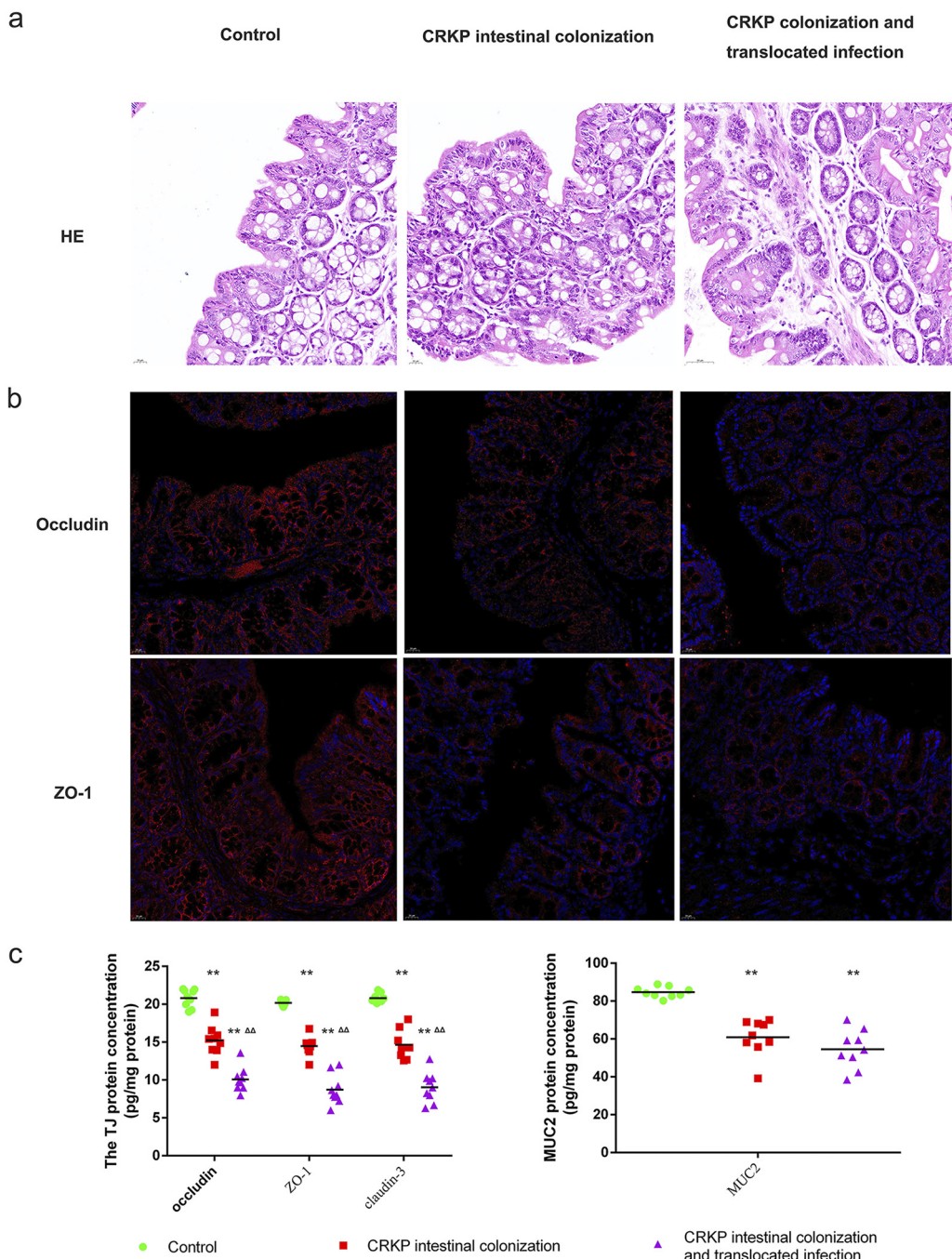

**FIG 4** CRKP colonization and translocated infection disrupt the intestinal barrier. To evaluate the influences of CRKP rectal colonization and translocated infection on the intestinal barrier, we performed HE and fluorescence staining for occludin and ZO-1. The contents of tight junction (TJ) proteins (claudin-3, occludin, and ZO-1) and mucin2 (MUC2) were further analyzed by an ELISA. (a) HE staining was used to evaluate histopathological injury in three groups. (b) Expression of tight junction (occludin and ZO-1) proteins. The nucleus is shown in blue by DAPI staining. On the cell membrane, the tight junction marker is shown in red. (c) Statistics of the expression of tight junction proteins (claudin-3, occludin, and ZO-1) and MUC2 in the three groups. (**, $P < 0.001$ versus the control group; $\Delta\Delta$, $P < 0.001$ versus the CRKP rectal colonization group).

tazobactam, coupled with weak activity against carbapenems (12, 13). Ceftazidime or aztreonam typically retains activity as long as the isolates do not produce AmpC or ESBL enzymes. Observing antimicrobial activity *in vitro*, screening for the coexpression of carbapenemase genes such as $bla_{NDM-1}$, and surveying the AmpC or ESBL enzymes in clinical practice support antibiotic therapy options for infections caused by isolates harboring

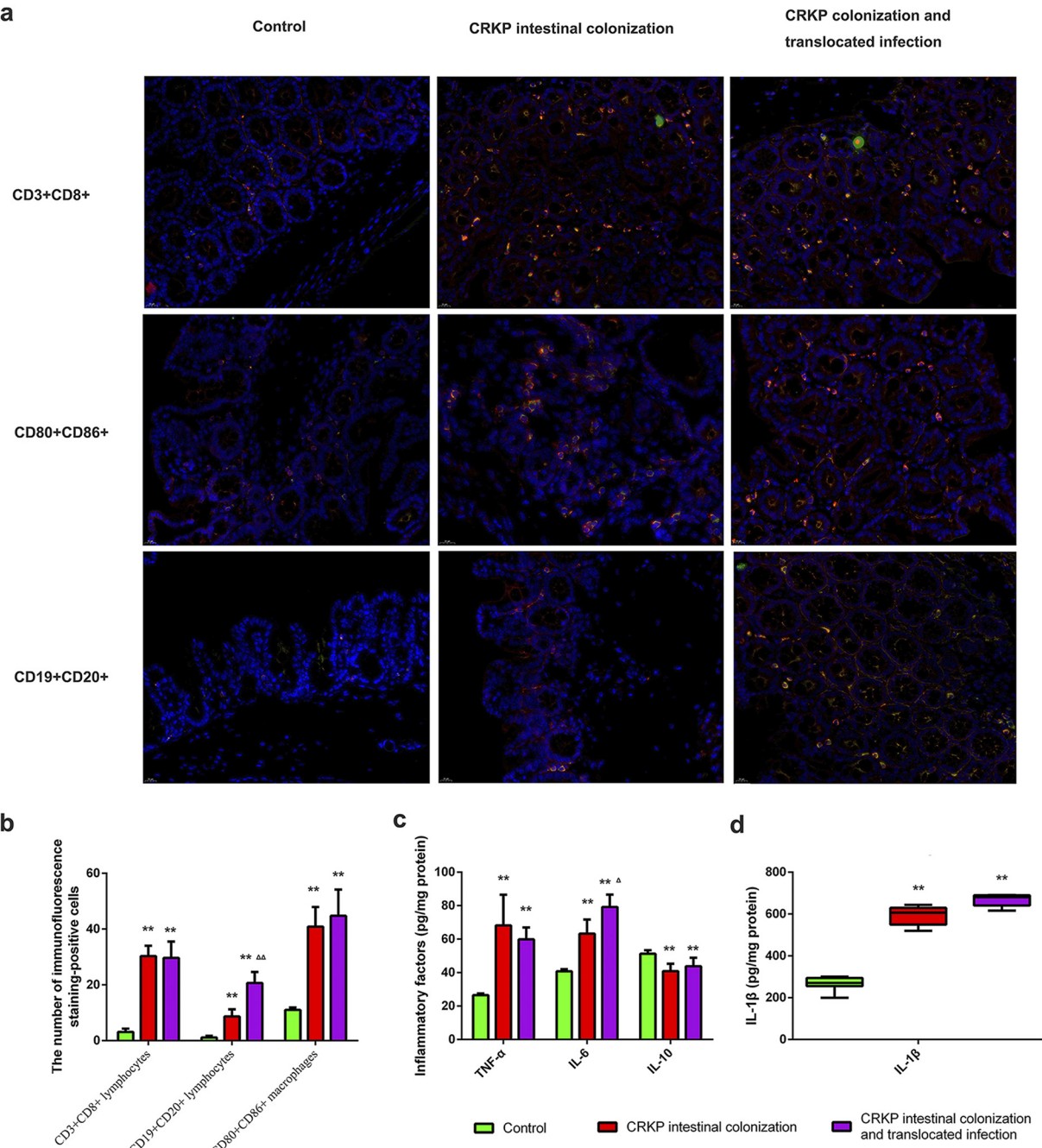

FIG 5 CRKP colonization and translocated infection induce obvious immune cell infiltration and inflammatory factor expression. To determine the features of immune cell infiltration in colon tissues subjected to CRKP colonization as well as translocated infection, double-immunofluorescence staining was performed in three groups of mice. We further evaluated the influence of CRKP colonization and translocated infection on inflammatory factor (IL-1$\beta$, TNF-$\alpha$, IL-6, and IL-10) expression. (a) Expression of CD3$^+$ CD8$^+$ lymphocytes, CD19$^+$ CD20$^+$ lymphocytes, and CD80$^+$ CD86$^+$ macrophages in colon tissues. By fluorescence microscopy, the nucleus is shown in blue with DAPI staining. Merged staining is shown in yellow/orange on the membrane, which reveals CD3$^+$ CD8$^+$ lymphocytes, CD19$^+$ CD20$^+$ lymphocytes, and CD80$^+$ CD86$^+$ macrophages. The numbers of double-immunofluorescence-stained positive cells (yellow/orange on the membrane) were captured by fluorescence microscopy. (b) Statistics of CD3$^+$ CD8$^+$ lymphocytes, CD19$^+$ CD20$^+$ lymphocytes, and CD80$^+$ CD86$^+$ macrophages in tissues harvested from animal models (**, $P < 0.001$ versus the control group; $\Delta\Delta$, $P < 0.001$ versus the CRKP intestinal colonization group). (c) Statistics of expression of inflammatory factors (IL-1$\beta$, TNF-$\alpha$, IL-6, and IL-10) in the three groups (**, $P < 0.001$ versus the control group; $\Delta$, $P < 0.05$ versus the CRKP rectal colonization group).

$bla_{OXA-48-like}$ genes. Although the mechanism for the dissemination of $bla_{OXA-48-like}$ genes is still unclear, our findings highlight the importance of the surveillance of $bla_{OXA-48-like}$ variants during rectal screening in high-risk patients in ICU wards.

Gastrointestinal colonization with potential pathogens is always a prerequisite for

the development of translocated infections. Here, this is the first study to directly observe translocated systematic infections in CRE intestinally colonized animal models. According to our records, 50% of rectally colonized patients probably develop subsequent CRE-associated pneumonia in the hospital, with a median time from admission to the onset of infection of 4 days. These patients with probable translocated pneumonia were significantly more likely to have a poor clinical outcome, with significantly high mortality rates. The identification of risk factors is important because it can help clinicians determine which patient needs rectal screening more. In the present study, we found that various factors were associated with subsequent pneumonia in patients with fecal colonization, including (i) enteral feeding, (ii) a nasogastric tube, (iii) carbapenem exposure, (iv) proton pump inhibitor (PPI) use, and (v) a total serum protein level of ≤60 g/L. Antibiotic pressure is a well-known determinant of the development and spread of antibiotic-resistant bacteria. Data from previous studies (1, 2, 27–29) have been very supportive of the association between CRE colonization or infection and exposure to antibiotics. The current study observed that the period after rectal carriage (median of 4 days) was the time frame in which exposure to antibiotics increased the risk of developing CRE infection. Regarding specific antibiotic classes, we identified exposure to carbapenems as being associated with subsequent CRE clinical infection. The clinical implication of our findings is that carbapenem therapy should be used with caution for CRE carriers, especially 2 to 11 days after gastrointestinal screening is positive for CRE colonization.

In addition to the role of carbapenem exposure, other findings in our study suggested that the transition from CRE rectal carriage to clinical infection was most likely to occur in the context of patients with enteral feeding. In patients with enteral feeding, fecal colonization has been shown to increase the risk of subsequent pneumonia more than 3-fold. Enteral feeding is required for nutritional support in patients with dysphagia. However, enteral feeding can inevitably lead to bacterial overgrowth and colonization of mucosal gut surfaces and can significantly diminish the proinflammatory cytokine response. Enteral feeding also increases the risks of reflux and changes in the pH or urea content (43). Several reports (33, 34) have shown that the intestinal microbiota composition was one of the factors responsible for the enhancement of intestinal permeability. To our knowledge, this is the first report to clarify that enteral feeding has additional effects on translocation from rectal carriage to clinical infection associated with CRE. At the same time, health care workers should comply with hand hygiene practices before preparing feeds for patients with nasogastric or percutaneous gastrostomy, especially for ill patients receiving concurrent carbapenem therapy and enteral feeding. Although the mechanism of enteral feeding and gastrointestinal colonization by CRE remains uncertain, the results of our experiment suggest that clinicians should be cautious when using enteral feeding as a therapeutic agent in critically ill patients with CRE rectal colonization. Consistent with the results of previous studies (26, 44–46), we found that the use of PPIs was also associated with translocated infection with CPE. Moreover, in the establishment of animal models of intestinal colonization, the use of PPIs was an indispensable prerequisite along with immunosuppressant dosing and antibiotic exposure. An explanation for this is that PPIs probably alter the composition of the gut microbiota and enhance intestinal permeability.

In this study, CRKP colonization significantly changed the composition of bacteria in feces, and the relative abundance of *Bacteroidota* decreased to 20.42%. The phylum *Firmicutes* also displayed a slightly decreased relative abundance compared with the control group. *Bacteroidetes* members are major acetate and propionate producers, while members of the phylum *Firmicutes* produce mostly butyrate. These downregulations are partly in agreement with SCFA results showing that propionate as well as butyrate were significantly decreased in fecal samples from CRKP-colonized patients. Acetate, propionate, and butyrate are the most common SCFAs, which, as the most abundant metabolites of the gut microbiome in the intestinal lumen, play important and multiangle roles in regulating the host's colonization resistance. The mucous layer provides the first physical barrier in the colon. MUC2 protects and limits the immunogenicity of intestinal antigens,

determines an anti-inflammatory state, and preserves features for the decolonization of pathogenic microorganisms. It is well known that goblet cells secrete gel-forming MUC2. As reported in a previous study (47), butyrate upregulates MUC2 expression by activating the MUC2 promoter and altering histone modifications in that DNA region. Specifically, a CRKP-colonized status significantly inhibited MUC2 protein, probably attributable to the downregulated SCFAs in this study. Moreover, CRKP intestinal colonization was further regarded as being the result of decreased MUC2 protein because GI colonization depends on the ability of the bacteria to adhere to mucosal surfaces to form a biofilm within the mucus layer, but whether the CRKP colonization profile is the cause or the consequence of downregulated MUC2 remains unclear. Alternatively, we also observed that *Klebsiella* and *Bacteroides fragilis* were the dominant taxa in the CRKP rectal colonization groups. Multiple investigations (48–50) have shown that antibiotic exposure causes dysbiosis, reduces colonization resistance by the stable resident microbiota, and promotes the expansion of pathogens. Furthermore, *Bacteroides fragilis* strains probably invade intestinal tissue and cause damage (51), which probably promotes CRKP colonization.

TJ proteins of enterocytes predominantly regulate the integrity of the intestinal barrier and play a vital role in bacterial translocated infection. In the present study, CRKP intestinal colonization treatment downregulated TJ protein expression, especially in mice with CRKP translocated infection. At the same time, CRKP colonization and translocated infection induced obvious immune cell infiltration and inflammatory factor expression. These complicated activities involve microbe-microbe and microbe-host interactions, which include complex bacterial networks, the host immune system, and mucus and intestinal epithelial barrier integrity. A number of studies (30–32, 35, 36) have concluded that butyrate is able to recover barrier function through the positive regulation of the expression of claudin-1, ZO-1, and occludin in cdx2-transformed IEC-6 cells (cdx2-IEC) and Caco-2 cells, resulting in increased transepithelial electrical resistance. Thus, we conclude that, to a certain extent, lower SCFA levels were probably associated with downregulated TJ proteins in CRKP-colonized mice. Our experiments also showed that CRKP colonization was associated with activated inflammation of the intestinal mucosa, which suggests that decreased beneficial bacteria and an overgrowth of CRKP probably induce gastrointestinal tract inflammation. Low-grade inflammation involves innate immune cells such as $CD80^+$ $CD86^+$ macrophages as well as adaptive immunity mediated by $CD3^+$ $CD8^+$ lymphocytes and $CD19^+$ $CD20^+$ lymphocytes, etc. Furthermore, regulatory T (Treg) cells likely participate in gastrointestinal tract inflammation through the expression of IL-10.

In CRKP translocated infection patients, the transition from CRKP rectal carriage to clinical infection was most likely to occur in the context of patients with lower serum protein levels. Our animal experiments showed that CRKP translocated infection did not obviously enhance the percentages of $CD3^+$ $CD8^+$ lymphocytes and $CD19^+$ $CD20^+$ lymphocytes in the CRKP-colonized colon. We and others (38) have suggested that gut microbiome alterations induce intestinal barrier dysfunction and thus allow bacterial translocation into the circulation, causing a strong systemic proinflammatory immune response. Interestingly, we observed that the enhanced $CD80^+$ $CD86^+$ macrophages in colon tissues derived from mice with translocated infection were in line with the lower levels of secretion of TNF-$\alpha$. In this regard, these data suggest that immune dysfunction is probably partly associated with CRKP translocated infection.

As part of a hostile takeover of the host niche, higher relative abundances in the intestinal flora are necessary before CRKP translated infection, and we will explore the mechanism involved in another investigation. One hypothesis is that CRKP translocated infection may affect metabolism disorders of the host and microbiome in CRKP rectally colonized patients. UPLC-MS/MS was used to analyze the fecal metabolic profiles in this study, which might provide opportunities for the identification of high-risk patients, intervention, and, ultimately, the prevention of infection. According to validated mathematical models, 13 altered metabolites in feces were obtained, which were involved mainly in amino acid metabolism, glucose metabolism, and lipid metabolism. Unfortunately, no correlation was observed between the 13 altered metabolites

in nontargeted metabolomes and the identified statistically significant microorganism biomarkers in the intestinal flora of CRKP translocated infection patients. However, downregulated propionate was observed in fecal samples collected from the CRKP translocated infection group compared with patients with only CRKP rectal colonization. Thus, we conclude that to a certain extent, in human feces, lower SCFA concentrations are associated with increased gut permeability, intestinal flora dysregulation, and proinflammatory immune features. Although microorganism biomarkers, such as *Streptococcus* and *Faecalibacterium*, and propionate were associated with CRKP translocation resistance, questions remain with respect to microbe-microbe and microbe-host interactions.

This study had several limitations. First, erroneous records or missing data in the medical records may have been present. Second, this study was performed at only one center, and other regions of Yunnan Province were not involved; thus, these results may not be completely representative of the epidemiology of CRE in China. Third, the sample size of patients colonized with CRE was relatively small. Studies with larger sample sizes are needed to verify the preliminary findings of the current study.

In summary, this study screened 54 patients hospitalized in the ICU with rectal colonization, and 50% of the colonized patients probably developed CRE-associated pneumonia, in line with the significantly high mortality rate. Moreover, the $bla_{\text{OXA-48-like}}$ gene has become the mainstream gene of CRE strains isolated by rectal screening, which are grouped into sequence type 231 in our geographical area of study. Rectal carriage of CRE with the $bla_{\text{OXA-48-like}}$ gene probably contributed to the increasing prevalence of translocated clinical infections. Here, our study found that the transition from CRE rectal carriage to clinical infection was most likely to occur in patients with carbapenem exposure and enteral feeding. In addition, the use of PPIs was also associated with translocated infection among CRE-colonized patients. We suggest that clinicians should be cautious when using enteral feeding as a therapeutic agent in critically ill patients with CRE rectal colonization, especially 2 to 11 days after gastrointestinal screening. Furthermore, CRKP colonization and translocated infection influenced the diversity and community composition of the intestinal microbiome. Downregulated propionate and butyrate probably play important and multiangle roles in regulating immune cell infiltration, inflammatory factor expression, and mucus and intestinal epithelial barrier integrity. Although the risk factors and intestinal biomarkers for subsequent infections among CRKP-colonized patients were explored, questions remain regarding how CRKP interacts with host intestinal epithelia and the mechanism of potential intestinal translocation. Further work is needed to elucidate the complicated mechanisms with respect to microbe-microbe and microbe-host interactions.

## MATERIALS AND METHODS

**Study design and sample collection.** This prospective surveillance study was done in a 30-bed ICU of a tertiary care hospital in Yunnan Province, China, between January and December 2019. Two rectal swabs (one each on day 1 and day 4 of ICU admission) were simultaneously collected during active surveillance from patients being admitted to the ICU ward, and inpatients were categorized into two groups based on rectal colonization: CRE colonization or noncolonization. Patients with positive cultures from rectal swabs but no symptoms of infection were included in the colonized group. Patients with rectal surveillance cultures that were continuously negative for CRE who were concurrently hospitalized in the same units as the colonized group of patients were evaluated as the control group. The exclusion criteria were being transferred and being discharged from the hospital within 48 h after the detection of colonization. Those patients with CRE colonization were followed up, and inpatients with both symptoms of pulmonary infection and positive cultures from the respiratory tract were defined as having subsequent pneumonia associated with CRE. Among patients with repeated growth of the same CRE isolate with an identical sensitivity profile, only the first isolate was included in the study. CRE-associated pneumonia was diagnosed by clinicians, and bronchoalveolar lavage as well as sputum suction fluids were studied to make the microbiological diagnosis.

**Data collection.** The clinical records of the patients in our study were collected and analyzed from patient charts and the electronic hospital database. The following demographic data were collected: age and gender. Data for the following clinical variables were collected: hospitalization history in the 6 months before screening; comorbid conditions (diabetes mellitus [DM]; cardiovascular, renal, lung, or neurological disease; malignancy; and immunodeficiency); surgery; mechanical ventilation; exposure to antibiotics; drug use (PPIs, hormones, and antifungal drugs); the presence of invasive devices, including a permanent urinary catheter, a central venous catheter (CVC), an enteral feeding tube, a drain, and an endotracheal tube; length of ICU stay;

APACHE II score; carriage of or infection by another multidrug-resistant organism (MDRO) (i.e., vancomycin-resistant enterococci [VRE], methicillin-resistant *Staphylococcus aureus* [MRSA], and ESBL-carrying organisms); and death associated with infection.

**Microbiological methods.** The CRE phenotype was defined using CDC criteria to identify *Enterobacteriaceae* as being nonsusceptible to imipenem, meropenem, or ertapenem. CarbaNP and mCIM with or without an eCIM test were further performed on all isolates to determine whether any bacteria produced carbapenemases by phenotypic methods but were negative by genotypic methods or vice versa.

**DNA extraction and gene amplification by PCR.** Genomic DNA from CRE species was prepared using the boiling method. The sequences of the primers used in this study are presented in Table S1 in the supplemental material. PCR was performed in a final volume of 20 $\mu$L containing 20 ng DNA, 0.47 $\mu$M each primer, 3 $\mu$L double-distilled water (ddH$_2$O), and 10 $\mu$L PCR mix buffer (TuoQin, China) using a thermal cycler (catalog number AGT9601; Bioanyu China). The positive PCR products were sequenced by TuoQin Biotechnology Co., Ltd. (Beijing, China), and the sequences were compared to the sequences in the GenBank database.

**Multilocus sequence typing.** Multilocus sequence typing (MLST) PCR with housekeeping genes was performed on all isolates according to protocols described on the Pasteur MLST websites (https://bigsdb.pasteur.fr/klebsiella/, https://pubmlst.org/bigsdb?db=pubmlst_ecloacae_seqdef, and https://bigsdb.pasteur.fr/ecoli/). The PCR product was sequenced, and the sequence was analyzed to determine the alleles and sequence types (STs) of the isolates.

**Fecal DNA extraction and 16S rRNA gene sequencing.** Stool samples were collected during the morning hours in a screw-cap collection container (DNA/RNA Shield fecal collection tube; Zymo Research, CA, USA). Stool specimens were stored at $-20°C$ until analysis. Fecal DNA was extracted using the cetyltrimethylammonium bromide (CTAB)-SDS method. 16S rRNA genes of distinct regions were amplified using specific primers. Sequencing of the 16S rRNA gene was performed using the 454 GS Junior system (Roche, Basel, Switzerland). Sequencing libraries were generated using a TruSeq DNA PCR-free sample preparation kit (Illumina, USA) according to the manufacturer's recommendations, and index codes were added. Library quality was assessed using the Qubit 2.0 fluorometer (Thermo Scientific) and the Agilent Bioanalyzer 2100 system. Microbiome bioinformatics analyses were performed using QIIME2 according to the official tutorials, with slight modifications. The Venn program was used to identify the overlap and uniqueness of OTUs in the two groups. Species annotation was performed using QIIME2 software and the Silva database. The biological diversity of the gut microbiota was estimated by the phylum, family, and genus compositions. Alpha diversity was applied for analyzing the complexity of species diversity for a sample through Chao1 and Shannon indices. Beta diversities on both weighted and unweighted UniFrac distances were calculated using QIIME2 software. The linear discriminant analysis (LDA) effect size (LEfSe) method (LDA score threshold of 4) was used for the quantitative analysis of biomarkers within different groups. This method was designed to analyze data for which the number of species or functional annotations is much higher than the number of samples and to provide biological class explanations to establish statistical significance, biological consistency, and effect size estimations for predicted biomarkers.

**Ultraperformance liquid chromatography-tandem mass spectrometry assay for nontargeted metabolomics.** Fecal samples were extracted by adding 400 $\mu$L of a methanol-acetonitrile-water (2:2:1, vol/vol/vol) solution to 100-mg fecal samples. After vortex mixing for 5 min, 3 freeze-thaw cycles ($-80°C$ for 30 min and 4°C for 10 min) were performed, followed by 10 cycles of ultrasonication. After centrifugation at 13,800 $\times$ *g* for 15 min at 4°C, the supernatant was measured by ultraperformance liquid chromatography-tandem mass spectrometry (UPLC-MS/MS). All data were acquired using MassLynx V4.1 (Waters Corporation, Milford, MA, USA) and imported into Progenesis QI V2.0 (Waters Corporation, Milford, MA, USA) to clean background noise, be normalized by a reference sample, correct the retention time, pick peaks, and identify compounds with databases.

**Short-chain fatty acid extraction and gas chromatography-mass spectrometry.** Short-chain fatty acids (SCFAs) were extracted from fecal samples according to methods described previously by Zhang et al. (52). All vials were stored at $-20°C$ before gas chromatography (GC) analysis. The GC-MS system consisted of a Thermo Trace 1300 system (Thermo Fisher Scientific, USA) and a Thermo ISQ 7000 system (Thermo Fisher Scientific, USA). Results were acquired using Chemstation software (Hewlett-Packard, Palo Alto, CA, USA).

**Mouse studies.** A CRKP clinical isolate expressing GFP was used in this study. A tetramethyl rhodamine isocyanate (TRITC)–D-Lys plasmid (Shengguang Biotechnology, Xiamen, China) was electroporated into electrocompetent *K. pneumoniae* cells. For the generation of fecal colonization, male C57BL/6J mice in the immunosuppression state (provided by cytarabine injected intravenously, at 440 mg/m$^2$) were supplemented with sterile water with or without levofloxacin (10 mg/kg of body weight) combined with pantoprazole (100 mg/kg). After 14 days of treatment, these mice were treated with CRKP via oral gavage at a dose of approximately 4 $\times$ 10$^8$ CFU per mouse. On day 7 after oral gavage treatment, fecal samples were collected from individual mice every day to determine rectal colonization by CRKP. Mice were euthanized and imaged using the Lumina series III IVIS (PerkinElmer, Waltham, MA, USA) at a wavelength of 660 nm/710 nm (excitation/emission) and an exposure time of 10 s. Inhalation anesthesia was induced by delivering 3.5 to 4.5% isoflurane in oxygen to mice via the respiratory system (nose cone) and then maintained at a concentration of 1 to 2% during the imaging procedure. When the *ex vivo* IVIS confirmed CRKP colonization of the gastrointestinal tract (on day 10 after oral gavage treatment) and subsequent infection (on approximately day 25 after oral gavage treatment), the animal model was successfully established, and the animals were divided into two groups (six mice per group): the CRKP intestinal colonization group and the CRKP translocated infection group. The control mice (*n* = 6) were treated with levofloxacin and pantoprazole only. Mice were sacrificed, and colon tissues were harvested and divided into parts for ELISAs and histological analyses.

**ELISA.** The concentrations of tight junction proteins and cytokines in colon tissue were measured using ELISA kits. Colonic TNF-$\alpha$, IL-6, IL-1$\beta$, and IL-17 quantitative ELISA kits (CusaBio, Wuhan, China) as well as commercially available occludin, ZO-1, and claudin-3 ELISA kits (Senbeijia Biotechnology, Nanjing, China) were used according to the manufacturers' instructions. The protein concentration was measured by the bicinchoninic acid (BCA) method using a BCA protein assay kit (Beyotime Biotechnology, Shanghai, China). The results were expressed as picograms per milligram of protein in the colon.

**Hematoxylin and eosin and immunofluorescence staining.** Paraffin-fixed tissue samples were cut into sections of a 5-$\mu$m thickness and stained with hematoxylin and eosin (HE) for histological examination. Immunofluorescence staining was further performed to evaluate the histomorphological expression features. Sections were incubated with primary antioccludin (1:50; Santa Cruz), anti-ZO-1 (1:50; Santa Cruz), anti-CD3 (1:150; Abcam)/anti-CD8 (1:100; Abcam), anti-CD80 (1: 50; Abcam)/anti-CD86 (1:150; Abcam), and anti-CD19 (1:30; Abcam)/anti-CD20 (1:30; Abcam) antibodies at 4°C overnight. After the sections were washed with phosphate-buffered saline (PBS), they were incubated with Cy3-labeled goat anti-rabbit IgG antibodies (red) (1:1,000; KPL) and DyLight 488-labeled goat anti-mouse/rat IgG antibodies (green) (1:1,000; Abcam) as the secondary antibodies. Finally, the sections were incubated with 4′,6-diamidino-2-phenylindole (DAPI) to stain the nuclei (blue) and observed using a fluorescence microscope (Olympus).

**Ethical considerations.** The data and samples analyzed in the present study were obtained in accordance with the standards and approval of The Affiliated Hospital of Yunnan University Biomedical Ethics Committee. The animal study was approved by the Animal Experimental Ethical Committee of Yunnan University.

**Statistical analysis.** GraphPad Prism 7 and SPSS 22.0 were used to analyze the data. A $P$ value of <0.05 was considered to indicate statistical significance. Univariate analyses were performed separately for each of the variables. Categorical variables were compared using a chi-square test. Continuous variables were compared using Student's $t$ test (normally distributed variables) and a Wilcoxon rank sum test (nonnormally distributed variables) as appropriate. The odds ratios (ORs) and their corresponding 95% confidence intervals (CIs) were calculated. Variables with $P$ values of <0.05 upon univariate analysis were evaluated as potential covariates in a stepwise multivariate logistic regression model. Microbiota-related analyses were conducted using QIIME and R 3.5.0. The LEfSe algorithm was performed using Python 2.7 and R 3.5.0. Network diagrams of samples and OTUs and bacterial interaction patterns of the validation cohort were created using Cytoscape 3.6.0.

## SUPPLEMENTAL MATERIAL

Supplemental material is available online only.

**SUPPLEMENTAL FILE 1**, PDF file, 0.2 MB.

## ACKNOWLEDGMENTS

We thank Jie Sun and Xianzhong Cheng (Intensive Care Union, The Affiliated Hospital of Yunnan University) for their assistance in the collection of APACHE II scores and clinical data.

This study was partially supported by the Association Foundation Program of the Yunnan Provincial Science and Technology Department and Kunming Medical University (grant number 202201AY070001-267) and by the Foundation of the Yunnan Health Training Project of High Level Talents (grant number H-2019046).

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
