## [Reviewer comments · Microbiology Spectrum]

Microbiology Spectrum

The clinical risk factors, microbiological and intestinal characteristics in Carbapenemase-producing Enterobacteriaceae colonization and subsequent infection

Wen-Li Yuan, Jia-Li Xu, Lin Guo, Yonghong Chen, Jinyi Gu, Huan Zhang, Chenghang Yang, Qiuping Yang, Shuwen Deng, Longlong Zhang, Qiong-Fang Deng, Zi Wang, Bin Ling, and Deyao Deng

Corresponding Author(s): Deyao Deng, The Affiliated Hospital of Yunnan University, The second hospital of Yunnan Province

Review Timeline:

Submission Date:	October 13, 2021
Editorial Decision:	January 3, 2022
Revision Received:	August 2, 2022
Accepted:	October 24, 2022

Editor: Arryn Craney

Reviewer(s): Disclosure of reviewer identity is with reference to reviewer comments included in decision letter(s). The following individuals involved in review of your submission have agreed to reveal their identity: Rafael Vignoli (Reviewer #2)

Transaction Report:

DOI: <https://doi.org/10.1128/Spectrum.01906-21>

January 3, 2022

Prof. Deyao Deng
The Affiliated Hospital of Yunnan University, The second hospital of Yunnan Province
Department of Clinical Laboratory
No. 176 Qing Nian Road Wuhua District,
Kunming, Yunnan Province 650021
China

Re: Spectrum01906-21 (Carbapenemase-producing Enterobacteriaceae colonization and subsequent infection in critically ill patients)

Dear Prof. Deyao Deng:

Unfortunately, the two peer reviewers both felt strongly that your manuscript requires major revisions before considering for publication.

- (1) Please revise the manuscript thoroughly for grammar/syntax and readability - many sentence are difficult to interpret.
- (2) Colonization with CREs is a well studied field, the reviewers felt that the clinical data and mouse work were disjointed.
- (3) It was also noted that not all beta-lactamase nomenclature/information is correct. Please review your manuscript to align with current literature. In addition, line 89, this statement is incorrect and the reference (#16) does not state the information referenced. The reference states "Furthermore, CAZ/AVI has no activity against class B carbapenemases, such as New Delhi metallo- β -lactamase (NDM)". Please correct.

Link Not Available

Sincerely,

Arryn Craney

Journals Department
Reviewer comments:

Reviewer #1 (Comments for the Author):

The Authors wrote a manuscript on carbapenemase-producing Enterobacteriaceae colonization and subsequent infection in critically ill patients. Clinical data regarding colonization status and clinical features of patients included in the study present limited originality. The study regarding colonization in mice using a model of CRKP expressing GFP is more original, but seems unconnected with clinical data in its current form.

Major concerns

- the use of the English language should be extensively reviewed
- the section regarding mice experiments is limited and should be expanded to better underline the novelty of the results.

Minor concerns

- please modify the carbapenemase genes names according to correct beta-lactamase nomenclature
- L82-83.VIM should be considered as major carbapenemase
- L103-105. Depends on the local epidemiology, OXA-48-like enzymes are not the most diffused worldwide.
- L130. Showing CRE colonization
- L131. Were followed up
- L215. *K. pneumoniae*
- L220-224. External primers should be used to determine the allelic variant of carbapenemase genes

Reviewer #2 (Comments for the Author):

The objective of the authors was to identify the independent risk factors for the Carbapenemase resistant enterobacteria (CRE) colonization and subsequently pneumonia in critically ill patients.

Carbapenemase production was studied by phenotypic methods and confirmed by PCR. The sequence-type of isolates from colonization and infection was determined by MLST. In addition, the authors performed a set of studies in a murine model.

While the paper presents interesting points, it has some major problems of design and data presentation that I discuss below:

Major or general comments:

The authors define the group of colonized patients as those with CRE detected by rectal swab on admission and as non-colonized those without CRE on admission. What about those who colonized after admission? These patients should also be included in the colonized group.

On the other hand, the authors assume that pneumonia occurs by bacterial translocation, but there are no studies of oro-pharyngeal colonization. The descending via could be possible as well. I would assume that bacterial translocation could be raised as a possibility in those cases where oro-pharyngeal colonization is not demonstrated. Throughout the paper, bacterial translocation should be treated as a possibility, not a certainty.

How many of the pneumonias are associated with mechanical ventilation?

The results are confusing and very difficult to follow: Lines 198 to 201 state that 681 patients were included in the study of which 54 belonged to the colonized group. Of the 54 colonized patients, 27 presented pneumonia produced by a CRE. However, Table S2 states that 320 patients were not colonized. Why were the remaining patients not included? The exclusion criteria should be included in materials and methods. On the other hand, it is not clear how many patients belonging to the non-colonized group presented respiratory symptoms due to CRE.

Line 236 states "The majority of CRE associated pneumonia (117, 57.6%) were hospital-acquired". Does this mean that there were 203 cases of CRE associated pneumoniae? How many of these were included in the study? Of the 117 hospital-acquired CRE associated pneumoniae cases, 27 belonged to the colonized group and 90 to the non-colonized? This information should be better presented.

The authors do not describe the results corresponding to the experiments in mice. In lines 252 to 256 the methodology is repeated but no results are included.

The criteria for referring to genes or proteins should be unified and spelled correctly, e.g. *bla_{KPC-2}*. All annotation even in tables contains errors.

Minor comments:

Title: The title is too general for what the paper describes. Considering that the only infections considered are respiratory and that colonization is only sought at ICU admission, the title should take these two aspects into account.

Lines 103 to 105: OXA-48 is not the most frequent carbapenemase worldwide, in the sentence it should be added to which geographical area it refers to.

Lines 123 to 131: The problem of defining colonized patients was previously discussed. Patients colonized by CRE once admitted to the ICU should also become part of the colonized group.

Lines 130 to 132: Given the nature of the study, the diagnostic criteria for defining pneumonia and how it is separated from, for example, purulent tracheobronchitis should be explained. Similarly, it should be better described which specimens were studied to make the microbiological diagnosis. Bronchoalveolar lavage? Sputum? How many of the pneumonia correspond to ventilator-associated pneumonia?

Lines 148 to 152:

Were antibiotic sensitivity studies performed on CRE isolates? For CDC criteria a bibliographic reference should be added, the same for CarbaNP, mCIM, etc.

Line 170 : Was the CRKP strain expressing GFP obtained commercially? Was it provided by a research group that has already published it? Was it obtained by the authors? If so, briefly describe how it was obtained experimentally.

How many mice were used?

Lines 177 to 181: The wording is confusing. As written it appears that the mice were first sacrificed and then given anesthesia to take the pictures.

Lines 219 to 222: This sentence should be condensed and corrected. The authors state that the predominant carbapenemase is OXA-48, however OXA-48 and KPC-2 are practically found in the same proportion, 24 and 23 isolates respectively. This interpretation is repeated for the clinical isolates analyzed below and also in the discussion and should not be put so conclusively. The predominant carbapenemases were both OXA-48 and KPC-2.

Line 224: the number of *Klebsiella pneumoniae* isolates producing blaKPC-2 is 22, so 20/23 should be corrected by 20/22.

Lines 227 to 230: Table 2 does not include the characteristics of the clinical isolates. While it is obvious that all blaOXA-48 like producers belong to ST231, it would be good to clarify to which ST the blaKPC-2 producers belong. Perhaps by putting the number in super index above each ST.

In Table 2: ST 131 and 88 seem to be interchanged.

Lines 236 to 241: All these data should be presented more consistently as previously stated. See previous comments.

Additionally the patient populations analyzed overlap (the 25/54 patients include the 17/27). The definition of clinical failure should be defined in materials and methods.

Lines 252 to 256: see previous comments, no description of results, or allusion to Figure 1.

Lines: 258 to 259: These percentages of rectal colonization by CRE are at ICU admission. Both aspects should be added: that it is rectal colonization and that it is on admission.

Lines 271 to 273: Without antibiotic susceptibility results it is not possible to assume that digestive decontamination could be effective. The authors could state this in a less categorical way and include bibliographic references in this regard. Not all digestive decontamination plans are equally effective.

Lines 280 to 282: A bibliographic reference should be included for this statement.

Line 282: The authors begin the sentence with "As in other studies...." Was the determination of SCFA levels in the intestine performed in this work? The entire following paragraph is speculative and does not follow from the results.

Line 291 to 299: These sentences do not seem to be consistent with each other. "As expected, this study observed that 23 CPE isolates harboring KPC-2 gene and clone ST231 and ST11 were the most prevalent STs. Why as expected? *Klebsiella pneumoniae* ST231 was an OXA-48 producer, why is KPC-2 and not OXA-48 mentioned in this sentence? But it immediately continues with "...we found that OXA-48-like gene was the predominant carbapenemase gene..." Please make a single sentence out of these two statements.

What is the meaning of this sentence: "OXA-48 variants have been recognized as the second or third most common CPE variant and rarely detected in tertiary care centers". Did the authors mean to say that OXA-48 is not detected in tertiary care centers? where?

But then they add "An explanation for this high rate was that those bacteria were isolated from rectal screening and not from other clinical sites" but the authors report isolates of respiratory origin as well (I don't understand the meaning of the sentence).

Lines 301 to 305: The statements are correct, but the work does not report the antibiotic susceptibility of the isolates, so it is not enough to guide therapy to know the type of carbapenemase they present. Are the isolates reported ESBL or ampC producers?

Line 310 to 312: This sentence seems to be based on results that were not included in the work.

Staff Comments:

Preparing Revision Guidelines

Please return the manuscript within 60 days; if you cannot complete the modification within this time period, please contact me. If you do not wish to modify the manuscript and prefer to submit it to another journal, please notify me of your decision immediately so that the manuscript may be formally withdrawn from consideration by Microbiology Spectrum.

Dear Editors,

Thank you very much for giving us the opportunity to revise our manuscript entitled “The clinical risk factors, microbiological and intestinal characteristics in Carbapenemase-producing Enterobacteriaceae colonization and subsequent infection” (the revised title) (Manuscript Number: Spectrum01906-21). We would like to thank you and the reviewers very much for positive and constructive comments and suggestions, which without doubt have helped us to improve our manuscript. We enclosed the revised manuscript, in which we believe that we have adhered to all instructions of *Microbiology Spectrum*. We have studied the reviewer’s comments carefully and have tried our best to revise our manuscript to meet their concerns, as you will see from the enclosed point-by-point response.

I look forward to hearing from you.

With best regards,

Deyao Deng

Department of Clinical Laboratory, The Affiliated Hospital of Yunnan University, The second hospital of Yunnan Province, Kunming, Yunnan Province, China.

No. 176 Qing Nian Road

Wuhua District, Kunming, Yunnan Province

P.R. China

650021

Tel: +86-871-65156650-2848

E-mail: dengdeyao2007@yeah.net

Editor

Comment 1: Please revise the manuscript thoroughly for grammar/syntax and readability - many sentence are difficult to interpret.

Response 1: We checked and corrected the spelling and syntax errors thoroughly. Furthermore, the entire manuscript has been re-edited by a native English speaker familiar with the topic.

Comment 2: Colonization with CREs is a well studied field, the reviewers felt that the clinical data and mouse work were disjointed.

Response 2: Our objective in the current study was to Identification of the risk factors and intestinal biomarkers for subsequent infections among CRE-colonized patients, which can be used to control those factors and to direct empirical antimicrobial therapy when necessary. We supplemented some experiments associated with gut microbiota diversity as well as feces metabolic profiles. The intestinal barrier, inflammatory factors and infiltrated immune cell were further investigated in colon tissues collected from CRKP colonized and translocated infection models. In conclusion, CRKP colonization and translocated infection influenced the diversity and community composition of the cecal microbiome. Down-regulated propionate and butyrate probably play an important and multiangle role in regulating immune cell infiltration, inflammatory factor expression, mucus and intestinal epithelial barrier integrity. We supplemented those and suggested that intestinal biomarkers, Streptococcus and Faecalibacterium, and propionate were associated with CRKP translocation resistant. Although the risk factors as well as biomarkers for subsequent infection among CRKP-colonized patients were explored, question remains how CRKP interacts with the host intestinal epitheliums and the mechanism of potential intestinal translocation. Further work is needed to elucidate the complicated mechanisms with respect to microbe-microbe and microbe-host interactions.

Comment 3: It was also noted that not all beta-lactamase nomenclature/information is correct.

Please review your manuscript to align with current literature. In addition, line 89, this statement is incorrect and the reference (#16) does not state the information referenced. The reference states "Furthermore, CAZ/AVI has no activity against class B carbapenemases, such as New Delhi metallo- β -lactamase (NDM)". Please correct.

Response 3: According to the consensus on β -Lactamase Nomenclature (Bradford PA, et al. 2022) and reviewer's comments, we modified the carbapenemase genes names in the proper format (italicized bla, followed by subscript allele designation) and corrected all beta-lactamase nomenclature/information. In addition, we corrected the reference (#16).

Reviewer 1

Major concerns

Comment 1: the use of the English language should be extensively reviewed

Response 1: We checked and corrected the spelling and syntax errors. Furthermore, the entire manuscript has been re-edited by a native English speaker familiar with the topic. We are sure the revised version of the manuscript will meet the concerns of the reviewers and requirements of the journal.

Comment 2: the section regarding mice experiments is limited and should expanded to better underline the novelty of the results.

Response 2: We appreciated the reviewer's comments for the suggestion, which are valuable in improving the quality of our manuscript. Colonization of the intestine by non-host niche microorganisms can lead to a permeable gut through various mechanisms, either being directly responsible for the feature of inflammation or favoring microbe-microbe and microbe-host interactions such as regulation the diversity and community composition of the cecal microbiome, manipulation bacterial metabolites, participating the innate and the adaptive immune response and penetration of host barriers, which allowing colonized microorganisms to invade the intestinal tissue and to subsequently translocated infection. In order to explore the complicated mechanisms, additional experiments were performed.

1. CRKP colonization and translocated infection influenced the diversity and community composition of the intestinal microbiome

1.1 To evaluate the effect of CRKP colonization and translocated infection on the diversity and community composition of the intestinal microbiome, the gut microbiota of patients in three groups were investigated based on 16S rRNA sequencing. Feces collected from CRKP colonization patients showed dramatic alteration of the gut flora composition compared with the control at the phylum level. Dominant communities of seven taxa and eleven taxa were found in the normal and CRKP rectal colonization groups, respectively.

1.2. In this experiment, we further detected the gut microbiota diversity between CRKP colonization group and translocated infection group. The result shown that *Klebsiella* (genus) was the dominant in the CRKP translocated infection group; *Streptococcus_salivarius* (species), *Streptococcus* (genus), *Streptococcaceae* (family) and *Faecalibacterium* (genus) were the dominant in the CRKP colonization group. Relative abundance of selected taxa showed that the abundance of *treptococcus_salivarius*, *Streptococcus*, *Streptococcaceae* and *Faecalibacterium* significantly decreased in CRKP translocated infection group, but the abundance of *Klebsiella* (genus) significantly increased

2. CRKP colonization and translocated infection influenced feces metabolic profiles

2.1 To investigate whether CRKP colonization and translocated infection induce feces metabolic disorder, UPLC-MS/MS was used to analyze non-targeted metabolomics in feces collected from CRKP rectal colonized patients, and 13 altered metabolites in feces were obtained.

2.2 This study further observed the SCFAs concentration difference among control group, CRKP rectal colonized patients and translocated infection group. Down-regulated propionate and butyrate were observed in feces collected from CRKP rectal colonized patients compared with control group, especially in CRKP translocated infection group.

3. CRKP colonization and translocated infection disrupted the intestinal barrier and induced the obvious immune cell infiltration and inflammatory factor expression in animal models.

3.1 To evaluate the influence of CRKP rectal colonization as well as the translocated infection on the intestinal barrier, tight junction (TJ) proteins and mucin2 (MUC2) were measured in

colon tissue by ELISA and immunofluorescence staining.

3.2 To determine the feature of immune cell infiltration in colon tissues treated with CRKP colonization as well as translocated infection, we performed immunofluorescence double staining of CD3⁺CD8⁺lymphocytes, CD19⁺CD20⁺lymphocytes, and CD80⁺CD86⁺macrophages in each group. Inflammatory factors (IL-1 β , TNF- α , IL-6 and IL-10) in the colon tissues of CRKP colonization-treated mice were also observed.

In conclusion, CRKP colonization and translocated infection influenced the diversity and community composition of the cecal microbiome. Down-regulated propionate and butyrate probably play an important and multiangle role in regulating immune cell infiltration, inflammatory factor expression, mucus and intestinal epithelial barrier integrity. We supplemented those and suggested that intestinal biomarkers, *Streptococcus* and *Faecalibacterium*, and propionate were associated with CRKP translocation resistant. Although the risk factors as well as intestinal biomarkers for subsequent infections among CRKP-colonized patients were explored, question remains how CRKP interacts with the host intestinal epitheliums and the mechanism of potential intestinal translocation. Further work is needed to elucidate the complicated mechanisms with respect to microbe-microbe and microbe-host interactions.

Minor concerns

Comment 1: please modify the carbapenemase genes names according to correct beta-lactamase nomenclature

Response 1: According to the consensus on β -Lactamase Nomenclature (Bradford PA, et al. 2022), we modified the carbapenemase genes names in the proper format (italicized bla, followed by subscript allele designation)

Comment 2: L82-83.VIM should be considered as major carbapenemase

Response 2: Thanks for the suggestion. We corrected in the manuscript.

Comment 3: L103-105. Depends on the local epidemiology, OXA-48-like enzymes are not the

most diffused worldwide.

Response 3: Thanks for the suggestion. We have revised them.

Comment 4: - L130. Showing CRE colonization,- L131. Were followed up

Response 4: Thanks for the suggestion. We have revised respectively.

Comment 5: - L220-224. External primers should be used to determine the allelic variant of carbapenemase genes.

Response 5: We appreciate your comment on the suggestion of external primers. In the current study, we investigated the fecal carriage of CRE, the carbapenemase genotypes, and identified the independent risk factors for the CRE colonization and subsequently translocated pneumonia in critically ill patients admitted to ICU from a university hospital in China. Intestinal flora composition and feces metabolic profiles were also observed. To our knowledge, there is scarce information about identified CRE rectal carriers is prone to have a subsequent infection with CRE. To test the hypothesis, we established gastrointestinal colonized animal models with CRKP clinical isolate expressing GFP in C57BL/6J mice and traced the subsequently system infection. The intestinal barrier, inflammatory factor and infiltrated immune cell were further investigated in colon tissues collected from CRKP colonized and translocated infection models. Our objective was to Identification of the risk factors and biomarkers for subsequent infections among CRE-colonized patients, which can be used to control those factors and to direct empirical antimicrobial therapy when necessary. Therefore, we can't complement external primers in the current paper. External primers will be used in our future work to determine the allelic variant of carbapenemase genes.

Reviewer 2

Major comments

Comment 1: The authors define the group of colonized patients as those with CRE detected by rectal swab on admission and as non-colonized those without CRE on admission. What about

those who colonized after admission? These patients should also be included in the colonized group.

Response 1: Thanks for the suggestion. In order to observe the prevalence of CRE colonization in hospitalized patients, those who colonized after admission should also be included in the colonized group,. However, we mainly focus on the microbiological characteristic, clinical risk factors and gut biomarkers in Carbapenemase-producing Enterobacteriaceae colonization and subsequent infection. To the best of our knowledge, our present study is the first report observed that microorganism biomarkers, Streptococcus and Faecalibacterium, and propionate were probably associated with CRKP translocation resistant. Hence, the occurrence seen in our investigation was deleted in the revised manuscript. At same time, we will complement our ongoing study design.

Comment 2: On the other hand, the authors assume that pneumonia occurs by bacterial translocation, but there are no studies of oro-pharyngeal colonization. The descending via could be possible as well. I would assume that bacterial translocation could be raised as a possibility in those cases where oro-pharyngeal colonization is not demonstrated. Throughout the paper, bacterial translocation should be treated as a possibility, not a certainty.

Response 2: The adult nasal microbiota differs between individuals, but species belonging to Corynebacterium, Propionibacterium, and Staphylococcus genera are the most abundant bacteria. In mammals, *K. pneumoniae* is a common species present in the gut. Many different environmental sources may be responsible for initial gastrointestinal colonization by *K. pneumoniae*. We and other investigations assume that pneumonia occurs by bacterial translocation was associated with intestinal colonization. Indeed, the descending via could be possible as well, however, oro-pharyngeal colonization was not demonstrated in our study. Hence, we revised the manuscript according to your suggestion.

Comment 3: How many of the pneumonias are associated with mechanical ventilation?

Response 3: Ventilator associated pneumonia (VAP), the most common and fatal nosocomial infection of critical care, is a new pneumonia that develops after 48 hours of endotracheal

intubation. Importantly, by the time of VAP onset, patients may have already been extubated. We reviewed the clinical records of the patients enrolled in our study. Of 54 patients with rectal CRE colonization, 4 cases were diagnosed pneumonias associated with mechanical ventilation. However, none CRE associated VAP was recorded. Our objective in the current study was to Identification of the risk factors and intestinal biomarkers for subsequent infections among CRE-colonized patients. VAP was not mainly concerned in the revised study.

Comment 4: The results are confusing and very difficult to follow: Lines 198 to 201 state that 681 patients were included in the study of which 54 belonged to the colonized group. Of the 54 colonized patients, 27 presented pneumonia produced by a CRE. However, Table S2 states that 320 patients were not colonized. Why were the remaining patients not included?

Response 4: Actually, 681 patients were included in the study of which 54 belonged to the colonized group, and the remaining patients (n=627) were concurrently hospitalized in the same units with continuous negative rectal surveillance cultures for CRE. Unfortunately, only 320 cases in non-colonized group evaluated with the APACHE II score by clinicians. Our objective in the current study was to Identification of the risk factors and intestinal biomarkers for subsequent infections among CRE-colonized patients. Hence, we revised the cases number to "374" and the percentages of rectal colonization by CRE in our investigation was deleted in the revised manuscript.

Comment 5: The exclusion criteria should be included in materials and methods. On the other hand, it is not clear how many patients belonging to the non-colonized group presented respiratory symptoms due to CRE.

Response 5: Thanks for the suggestion. The exclusion criteria were supplemented in the revised manuscript.

Comment 6: Line 236 states "The majority of CRE associated pneumonia (117, 57.6%) were hospital-acquired". Does this mean that there were 203 cases of CRE associated pneumoniae? How many of these were included in the study? Of the 117 hospital-acquired CRE associated

pneumoniae cases, 27 belonged to the colonized group and 90 to the non-colonized? This information should be better presented.

Response 6: I am sorry for the confused description. Actually, there were 117 cases of CRE associated pneumoniae by reviewing the clinical records. Of the 117 CRE associated pneumoniae cases in the study, 27 belonged to the CRE colonized group and 90 to the non-colonized group. Our objective in the study was to Identification of the risk factors and intestinal biomarkers for subsequent infections among CRE-colonized patients. Hence, VAP was not mainly concerned in the revised study. Furthermore, we want to state CRE associated HAP in a less categorical way.

Comment 7: The authors do not describe the results corresponding to the experiments in mice. In lines 252 to 256 the methodology is repeated but no results are included.

Response 7: Thanks for the suggestion. Our objective in the current study was to Identification of the risk factors and intestinal biomarkers for subsequent infections among CRE-colonized patients can be used to control those factors and to direct empirical antimicrobial therapy when necessary. We supplemented some experiments associated with gut microbiota diversity as well as feces metabolic profiles. The intestinal barrier, inflammatory factor and infiltrated immune cell were further investigated in colon tissues collected from CRKP colonized and translocated infection models. In conclusion, CRKP colonization and translocated infection influenced the diversity and community composition of the cecal microbiome. Down-regulated propionate and butyrate probably play an important and multiangle role in regulating immune cell infiltration, inflammatory factor expression, mucus and intestinal epithelial barrier integrity. We suggested that intestinal biomarkers, Streptococcus and Faecalibacterium, and propionate were associated with CRKP translocation resistant. Although the risk factors as well as biomarkers for subsequent infections among CRKP-colonized patients were explored, question remains how CRKP interacts with the host intestinal epitheliums and the mechanism of potential intestinal translocation. Further work is needed to elucidate the complicated mechanisms with respect to microbe-microbe and microbe-host interactions.

Comment 8: The criteria for referring to genes or proteins should be unified and spelled correctly, e.g. blaKPC-2. All annotation even in tables contains errors.

Response 8: Thanks for the suggestion. According to the consensus on β -Lactamase Nomenclature (Bradford PA, et al. 2022), we modified the carbapenemase genes names in the proper format (italicized bla, followed by subscript allele designation)

Minor comments:

Comment 1: Title: The title is too general for what the paper describes. Considering that the only infections considered are respiratory and that colonization is only sought at ICU admission, the title should take these two aspects into account.

Response 1: Our objective in the current study was to Identification of the risk factors and intestinal biomarkers for subsequent infections among CRE-colonized patients, which can be used to control those factors and to direct empirical antimicrobial therapy when necessary. So, we renamed the tittle as “The clinical risk factors, microbiological and intestinal characteristics in Carbapenemase-producing Enterobacteriaceae colonization and subsequent infection”

Comment 2: Lines 103 to 105: OXA-48 is not the most frequent carbapenemase worldwide, in the sentence it should be added to which geographical area it refers to.

Response 2: Thanks for the suggestion. We have revised them.

Comment 3: Lines 123 to 131: The problem of defining colonized patients was previously discussed. Patients colonized by CRE once admitted to the ICU should also become part of the colonized group.

Response 3: Thanks for the suggestion and we have revised them.

Comment 4: Lines 130 to 132: Given the nature of the study, the diagnostic criteria for defining pneumonia and how it is separated from, for example, purulent tracheobronchitis should be explained. Similarly, it should be better described which specimens were studied to make the microbiological diagnosis. Bronchoalveolar lavage? Sputum?

Response 4: The CRE associated pneumonia was diagnosed by clinicians, and bronchoalveolar lavage as well as sputum suction was studied to make the microbiological diagnosis in our study.

Comment 5: Lines 148 to 152: Were antibiotic sensitivity studies performed on CRE isolates? For CDC criteria a bibliographic reference should be added, the same for CarbaNP, mCIM, etc.

Response 5: Thanks for the suggestion. Antibiotic sensitivity studies were previously performed on CRE isolates, In the current study, our objective was to Identification of the risk factors and intestinal biomarkers for subsequent infections among CRE-colonized patients, which can be used to control those factors and to direct empirical antimicrobial therapy when necessary. So, data was not shown in the revised manuscript.

Comment 6: Line 170 : Was the CRKP strain expressing GFP obtained commercially? Was it provided by a research group that has already published it? Was it obtained by the authors? If so, briefly describe how it was obtained experimentally. How many mice were used?

Response 6: We generated the CRKP strain expressing GFP in the study, and briefly describe was supplemented in the revised manuscript.

Comment 7: Lines 177 to 181: The wording is confusing. As written it appears that the mice were first sacrificed and then given anesthesia to take the pictures.

Response 7: Thanks for the suggestion and we have revised them.

Comment 8: Lines 219 to 222: This sentence should be condensed and corrected. The authors state that the predominant carbapenemase is OXA-48, however OXA-48 and KPC-2 are practically found in the same proportion, 24 and 23 isolates respectively. This interpretation is repeated for the clinical isolates analyzed below and also in the discussion and should not be put so conclusively. The predominant carbapenemases were both OXA-48 and KPC-2.

Response 8: Thanks for the kind help. We have revised them according to your suggestion.

Comment 9: Line 224: the number of *Klebsiella pneumoniae* isolates producing blaKPC-2 is 22, so 20/23 should be corrected by 20/22.

Response 9: Thanks for the suggestion and we have revised them.

Comment 10: Lines 227 to 230: Table 2 does not include the characteristics of the clinical isolates. While it is obvious that all blaOXA-48 like producers belong to ST231, it would be good to clarify to which ST the blaKPC-2 producers belong. Perhaps by putting the number in super index above each ST. In Table 2: ST 131 and 88 seem to be interchanged.

Response 10: Thanks for the suggestion and we have revised them accordingly.

Comment 11: Lines 236 to 241: All these data should be presented more consistently as previously stated. See previous comments. Additionally the patient populations analyzed overlap (the 25/54 patients include the 17/27). The definition of clinical failure should be defined in materials and methods.

Response 11: Thanks for the suggestion and we have revised these data in the current manuscript. Actually, the patient populations analyzed overlap (the 25/54 patients include the 17/27). However, intestinal colonization with potential pathogens is always a prerequisite for the development of translocated infections, and the overlap was inevitable to a certain extent. Nevertheless, the sample size of colonized patients with CRE was relatively small. Studies of larger sample size are needed to verify the preliminary findings in the current study.

Comment 12: Lines 252 to 256: see previous comments, no description of results, or allusion to Figure 1.

Response 12: Thanks for the suggestion and we have revised as previous response.

Comment 13: Lines: 258 to 259: These percentages of rectal colonization by CRE are at ICU admission. Both aspects should be added: that it is rectal colonization and that it is on admission.

Response 13: Thanks for the suggestion. As shown in the response 1 to major comment, the percentages of rectal colonization by CRE in our investigation was deleted in the revised manuscript. At same time, we will complement our ongoing study design.

Comment 14: Lines 271 to 273: Without antibiotic susceptibility results it is not possible to assume that digestive decontamination could be effective. The authors could state this in a less categorical way and include bibliographic references in this regard. Not all digestive decontamination plans are equally effective.

Response 14: Thanks for the help and we have revised according to your suggestion.

Comment 15: Lines 280 to 282: A bibliographic reference should be included for this statement.

Response 15: Thanks for the suggestion and a bibliographic reference was included for the statement in the revised manuscript.

Comment 16: Line 282: The authors begin the sentence with "As in other studies...." Was the determination of SCFA levels in the intestine performed in this work? The entire following paragraph is speculative and does not follow from the results.

Response 16: To investigate whether CRKP colonization and translocated infection induce feces metabolic disorder, the study further observed the SCFAs concentration difference among control group, CRKP rectal colonized patients and translocated infection group. Down-regulated propionate and butyrate were observed in feces collected from CRKP rectal colonized patients compared with control group, especially in CRKP translocated infection group.

Comment 17: Line 291 to 299: These sentences do not seem to be consistent with each other. "As expected, this study observed that 23 CPE isolates harboring KPC-2 gene and clone ST231 and ST11 were the most prevalent STs. Why as expected? Klebsiella pneumonia ST231 was an OXA-48 producer, why is KPC-2 and not OXA-48 mentioned in this sentence?"

But it immediately continues with "...we found that OXA-48-like gene was the predominant carbapenemase gene..." Please make a single sentence out of these two statements.

Response 17: Thanks for the help and we have revised according to your suggestion.

Comment 18: What is the meaning of this sentence: "OXA-48 variants have been recognized as the second or third most common CPE variant and rarely detected in tertiary care centers".

Did the authors mean to say that OXA-48 is not detected in tertiary care centers? where?

Response 18: Thanks for the suggestion and the uncorrected describe was deleted.

Comment 19: But then they add "An explanation for this high rate was that those bacteria were isolated from rectal screening and not from other clinical sites" but the authors report isolates of respiratory origin as well (I don't understand the meaning of the sentence).

Response 19: I am sorry for the confusing describe. As a matter of fact, 51 strains harbouring carbapenemase genes screened in this study were isolated from rectal, and *bla*_{OXA-48-like} gene (24/51, 47%) was the predominant carbapenemase gene.

Comment 20: Lines 301 to 305: The statements are correct, but the work does not report the antibiotic susceptibility of the isolates, so it is not enough to guide therapy to know the type of carbapenemase they present. Are the isolates reported ESBL or ampC producers?

Response 20: Thanks for the suggestion and we have revised in the manuscript. Results obtained by Díaz-Agero Pérez C et al found that Local prevalence of ESBL producing Enterobacteriaceae intestinal carriers at admission and co-expression of ESBL and OXA-48 carbapenemase in *Klebsiella pneumoniae*. Hence, Observing the antimicrobial activity in vitro, screening co-expression carbapenemase gene, such as *bla*_{NDM-1}, and surveying the AmpC or ESBLs enzymes in clinical practice supports antibiotic therapy options for infections caused by isolates harboring *bla*_{OXA-48-like} genes.

Comment 21: Line 310 to 312: This sentence seems to be based on results that were not

included in the work.

Response 21: Thanks for the suggestion and the uncorrected describe was deleted.

October 24, 2022

Prof. Deyao Deng
The Affiliated Hospital of Yunnan University, The second hospital of Yunnan Province
Department of Clinical Laboratory
No. 176 Qing Nian Road Wuhua District,
Kunming, Yunnan Province 650021
China

Re: Spectrum01906-21R1 (The clinical risk factors, microbiological and intestinal characteristics in Carbapenemase-producing Enterobacteriaceae colonization and subsequent infection)

Dear Prof. Deyao Deng:

Thank you for revising your manuscript! The edits greatly strengthened the paper and I look forward to seeing it published.

Your manuscript has been accepted, and I am forwarding it to the ASM Journals Department for publication. You will be notified when your proofs are ready to be viewed.

Sincerely,

Arryn Craney
Editor, Microbiology Spectrum

Journals Department
Supplemental table1,2: Accept